# A Dynamical Analysis and New Traveling Wave Solution of the Fractional Coupled Konopelchenko–Dubrovsky Model

Jin Wang and Zhao Li *

College of Computer Science, Chengdu University, Chengdu 610106, China; wj19882009@163.com
* Correspondence: clizhao10.26@163.com

**Abstract:** The main object of this paper is to study the traveling wave solutions of the fractional coupled Konopelchenko–Dubrovsky model by using the complete discriminant system method of polynomials. Firstly, the fractional coupled Konopelchenko–Dubrovsky model is simplified into nonlinear ordinary differential equations by using the traveling wave transformation. Secondly, the trigonometric function solutions, rational function solutions, solitary wave solutions and the elliptic function solutions of the fractional coupled Konopelchenko–Dubrovsky model are derived by means of the polynomial complete discriminant system method. Moreover, a two-dimensional phase portrait is drawn. Finally, a 3D-diagram and a 2D-diagram of the fractional coupled Konopelchenko–Dubrovsky model are plotted in Maple 2022 software.

**Keywords:** Konopelchenko–Dubrovsky model; complete discriminant system; traveling wave solution; phase portrait

## 1. Introduction

In the present era, nonlinear evolution equations (NLEEs) [1–4] are employed in a number of areas like physics, chemistry, biology, fluid dynamics, engineering, optical fibers, plasma, and hydrodynamics. The analytical solutions of NLEEs can be applied to control complex behavior and difficult phenomena when the system displays [5–23]. While there is no unified method to obtain the exact solution of nonlinear evolution equations, most of the time, NLEEs can be converted into an ordinary differential equation by taken the traveling wave transformation. Based on the efforts of many predecessors, various methods have been imposed to solve NLEEs precisely and analyze various wave phenomena. He and Wu [24] proposed the first-time Exp-function method to seek solitary solutions, periodic solutions and compacton-like solutions of the KdV equation and Dodd–Bullough–Mikhailov equation. By using Hirota's bilinear transformation method, Ma proved the existence of N-soliton solutions of the (2+1)-dimensional KdV equation, the Kadomtsev–Petviashvili equation, the (2+1)-dimensional Hirota–Satsuma–Ito equation, and a combined pKP–BKP equation [25,26], respectively. Li et al. presented the $(w/g)$-expansion method [27]. Later, Arafat and his collaborators applied the customized $(w/g)$-expansion method to found the optical soliton solutions of the paraxial nonlinear Schrödinger equation and fractional Biswas–Arshed model [28,29], respectively. Wazwaz has derived the solitons and periodic solution of the Dodd–Bullough–Mikhailov and the Tzitzeica–Dodd–Bullough equations via the tanh-function approach [30]. Seadawy and Iqbal analyzed the nonlinear damped Korteweg–de Vries equation in an unmagnetized collisional dusty plasma via the direct algebraic approach [31]. Arafat also investigated scores of broad-spectral soliton solutions to the stated system via the auxiliary equation technique [32]. However, research on the traveling wave solutions of more complex fractional order NLEEs is still ongoing, and there are still a large number of open issues that need to be addressed by researchers.

In this paper, we consider the fractional coupled Konopelchenko–Dubrovsky model [33]

$$\begin{cases} D_t^\kappa u - u_{xxx} - 6auu_x + \frac{3}{2}b^2u^2u_x - 3v_y + 3bu_xv = 0, \\ v_x = u_y, \end{cases} \quad (1)$$

where $D_t^\kappa u$ is the conformable fractional derivative. $u = u(x,y,t)$ and $v = v(x,y,t)$ represent the velocity components along the horizontal and vertical axes, respectively. $a$ and $b$ stand for the amplitude of the wave. When $\kappa = 1$, Equation (1) become the well-known integer-order Konopelchenko–Dubrovsky model [34]. The main object of this paper is to study the traveling wave solution of the fractional coupled Konopelchenko–Dubrovsky model by using the complete discriminant system method of polynomials. On the one hand, the main effort of this article is to focus on constructing the traveling wave solution of Equation (1). On the other hand, without solving Equation (1), its dynamic branch will be analyzed.

The conformable fractional derivative was first proposed by Khalil et al. [35]. Compared with traditional fractional derivatives, the conformable fractional derivative has a more intuitive physical meaning. At present, it has been widely used in the construction of infectious disease dynamics models, nonlinear system modeling, and thermal science fields. Its definition is usually described as follows.

**Definition 1** ([36]). *Let $f : [0,\infty) \to \mathbf{R}$. Then, the conformable derivative of f of order $\kappa$ is defined as*

$$D_t^\kappa f(t) = \lim_{\varepsilon \to 0} \frac{f(t + \varepsilon t^{1-\kappa}) - f(t)}{\varepsilon}, \forall t \in (0, +\infty), \kappa \in (0,1], \quad (2)$$

*and the function f is $\kappa$-conformable differentiable at a point t if the limit in Equation (2) exists.*

The remaining sections of this article are arranged as follows: In Section 2, the traveling wave solutions of Equation (1) are constructed by using the complete discriminant system method. Moreover, a two-dimensional phase portrait is drawn. In Section 3, the three-dimensional, two-dimensional, and density plots to some obtained solutions of Equation (1) are plotted. Finally, a brief summary is presented.

## 2. Dynamical Analysis and Traveling Wave Solutions of Equation (1)

### 2.1. Traveling Wave Transformation

In this section, we first consider the wave transformation

$$u(x,y,t) = U(\xi), v(x,y,t) = V(\xi), \xi = kx + ly + \mu\frac{t^\kappa}{\kappa}, \quad (3)$$

Substituting Equation (3) into Equation (1), we have

$$\begin{cases} \mu U' - k^3U''' - 6akUU' + \frac{3}{2}kb^2U^2U' - 3lV' + 3kbU'V = 0, \\ kV' = lU'. \end{cases} \quad (4)$$

Integrating the second equation of Equation (4), we obtain

$$V = \frac{l}{k}U. \quad (5)$$

Substituting Equation (5) into the first equation of Equation (4), we have

$$-k^3U'' + \frac{1}{2}kb^2U^3 + (\frac{3lb}{2} - 3ak)U^2 + (\mu - \frac{3l^2}{k})U = d_1, \quad (6)$$

where $d_1$ is the integral constant.

### 2.2. Dynamical Analysis

Here, we consider the planar dynamic system of Equation (6) when $d_1 = 0$

$$\begin{cases} \frac{dU}{d\xi} = z, \\ \frac{dz}{d\xi} = \ell_3 U^3 + \ell_2 U^2 + \ell_1 U, \end{cases} \tag{7}$$

where $\ell_3 = \frac{b^2}{2k^2}$, $\ell_2 = \frac{1}{k^3}\left(\frac{3lb}{2} - 3ak\right)$, $\ell_1 = \frac{1}{k^3}\left(\mu - \frac{3l^2}{k}\right)$.

The first integration of Equation (7) is

$$H(U, z) = \frac{1}{2}z^2 - \frac{\ell_3}{4}U^4 - \frac{\ell_2}{3}U^3 - \frac{\ell_1}{2}U^2 = h. \tag{8}$$

By setting the parameter values of fixed Equation (7), we draw the planar phase portrait of Equation (7), as shown in Figure 1.

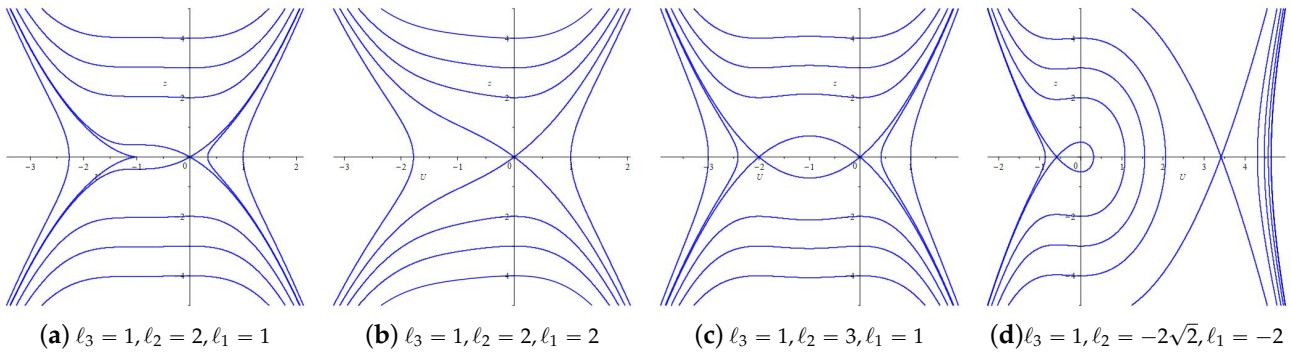

**(a)** $\ell_3 = 1, \ell_2 = 2, \ell_1 = 1$    **(b)** $\ell_3 = 1, \ell_2 = 2, \ell_1 = 2$    **(c)** $\ell_3 = 1, \ell_2 = 3, \ell_1 = 1$    **(d)** $\ell_3 = 1, \ell_2 = -2\sqrt{2}, \ell_1 = -2$

**Figure 1.** Phase portrait of Equation (7).

### 2.3. Traveling Wave Solutions of Equation (1)

Multiplying both sides of Equation (6) by $U'$ simultaneously and integrating it yields

$$(U')^2 = b_4 U^4 + b_3 U^3 + b_2 U^2 + b_1 U + b_0, \tag{9}$$

where $b_4 = \frac{b^2}{4k^2}$, $b_3 = \frac{lb}{k^3} - \frac{2a}{k^2}$, $b_2 = \frac{\mu}{k^3} - \frac{3l^2}{k^4}$, $b_1 = -\frac{2d_1}{k^3}$, $b_0 = 2d_2$; here, $d_2$ is the integral constant.

Here, we make a transformation:

$$\begin{cases} w = (b_4)^{\frac{1}{4}}\left(U + \frac{b_3}{4b_4}\right), \\ \chi = (b_4)^{\frac{1}{4}}\xi. \end{cases} \tag{10}$$

Substituting Equation (10) into Equation (9), we obtain:

$$w_\chi^2 = w^4 + c_2 w^2 + c_1 w + c_0, \tag{11}$$

where $c_2 = \frac{b_2}{\sqrt{b_4}}$, $c_1 = \left(\frac{b_3^3}{8b_4^2} - \frac{b_2 b_3}{2b_4} + b_1\right)(b_4)^{-\frac{1}{4}}$, $c_0 = \frac{-3b_3^4}{256b_4^3} + \frac{b_2 b_3^2}{16b_4^2} - \frac{b_1 b_3}{4b_4} + b_0$.

By integrating Equation (11) once, one has:

$$\pm(\chi - \chi_0) = \int \frac{dw}{\sqrt{w^4 + c_2 w^2 + c_1 w + c_0}}, \tag{12}$$

where $\chi_0$ denotes an integrating constant.

Denote that $G(w) = w^4 + c_2 w^2 + c_1 w + c_0$. We derive its complete discrimination system in the following form:

$$D_1 = 4, D_2 = -c_2, D_3 = -2c_2^3 + 8c_2c_0 - 9c_1^2, D_4 = -c_2^3c_1^2 + 4c_2^4c_0 + 36c_2c_1^2c_0 - 32c_2^2c_0^2 - \frac{27}{4}c_1^4 + 64c_0^3,$$
$$E_2 = c_1c_2^2 - 32c_2c_0. \tag{13}$$

According to the root-classifications of Equation (11), we will discuss the traveling wave solutions of Equation (1) under nine cases.

**Case 1:** $D_2 < 0, D_3 = 0, D_4 = 0, G(w) = [w^2 + q^2]^2$.

By combining Equation (9) with Equation (3), the traveling wave solutions of Equation (1) can be expressed as below:

$$u_1(x,y,t) = \frac{2ak - lb}{kb^2} + \frac{\sqrt{2(\mu k - 3l^2)}}{bk} \tan\left[\frac{\sqrt{2(\mu k - 3l^2)}}{bk^3}(\sqrt{\frac{b}{2k}}(kx + ly + \mu\frac{t^\kappa}{\kappa}) - \chi_0)\right]. \tag{14}$$

$$v_1(x,y,t) = \frac{2akl - l^2b}{k^2b^2} + \frac{l\sqrt{2(\mu k - 3l^2)}}{bk^2} \tan\left[\frac{\sqrt{2(\mu k - 3l^2)}}{bk^3}(\sqrt{\frac{b}{2k}}(kx + ly + \mu\frac{t^\kappa}{\kappa}) - \chi_0)\right]. \tag{15}$$

**Case 2:** $D_2 = 0, D_3 = 0, D_4 = 0, G(w) = w^4$. In this situation, the traveling wave solutions of Equation (1) take the form:

$$u_2(x,y,t) = \frac{2ak - lb}{kb^2} - \sqrt{\frac{2k}{b}}\left(\sqrt{\frac{b}{2k}}(kx + ly + \mu\frac{t^\kappa}{\kappa}) - \chi_0\right)^{-1}. \tag{16}$$

$$v_2(x,y,t) = \frac{2akl - l^2b}{k^2b^2} - l\sqrt{\frac{2}{bk}}\left(\sqrt{\frac{b}{2k}}(kx + ly + \mu\frac{t^\kappa}{\kappa}) - \chi_0\right)^{-1}. \tag{17}$$

**Case 3:** $D_2 > 0, D_3 = 0, D_4 = 0, G(w) = (w - \alpha)^2(w - \beta)^2$, where $\alpha$, $\beta$ are real number, and $\alpha > \beta$.

(i)   If $w > \alpha$ or $w < \beta$, the traveling wave solutions of Equation (1) take the form:

$$u_{3,1}(x,y,t) = \alpha\sqrt{\frac{k}{2b}} - \frac{lb - 2ak}{kb^2} + (\beta - \alpha)\sqrt{\frac{k}{2b}} \coth \frac{(\alpha - \beta)\left(\sqrt{\frac{b}{2k}}(kx + ly + \mu\frac{t^\kappa}{\kappa}) - \chi_0\right)}{2}. \tag{18}$$

$$v_{3,1}(x,y,t) = \frac{l\alpha}{\sqrt{2bk}} - \frac{l^2b - 2akl}{k^2b^2} + \frac{l(\beta - \alpha)}{\sqrt{2bk}} \coth \frac{(\alpha - \beta)\left(\sqrt{\frac{b}{2k}}(kx + ly + \mu\frac{t^\kappa}{\kappa}) - \chi_0\right)}{2}. \tag{19}$$

(ii)   If $\beta < w < \alpha$, the traveling wave solutions of Equation (1) take the form:

$$u_{3,2}(x,y,t) = \alpha\sqrt{\frac{k}{2b}} - \frac{lb - 2ak}{kb^2} + (\beta - \alpha)\sqrt{\frac{k}{2b}} \tanh \frac{(\alpha - \beta)\left(\sqrt{\frac{b}{2k}}(kx + ly + \mu\frac{t^\kappa}{\kappa}) - \chi_0\right)}{2}. \tag{20}$$

$$v_{3,2}(x,y,t) = \frac{l\alpha}{\sqrt{2bk}} - \frac{l^2b - 2akl}{k^2b^2} + \frac{l(\beta - \alpha)}{\sqrt{2bk}} \tanh \frac{(\alpha - \beta)\left(\sqrt{\frac{b}{2k}}(kx + ly + \mu\frac{t^\kappa}{\kappa}) - \chi_0\right)}{2}. \tag{21}$$

**Case 4:** $D_2 > 0, D_3 > 0, D_4 = 0, G(w) = (w - \alpha)^2(w - \beta)(w - \gamma)$, where $\alpha, \beta$ and $\gamma$ are real number, $\beta > \gamma$.

(i)   If $\alpha > \beta$ and $w > \beta$, or $\alpha < \gamma$ and $w < \gamma$, the implicit traveling wave solutions of Equation (1) can be expressed as below:

$$\pm\left(\sqrt{\frac{b}{2k}}(kx + ly + \mu\frac{t^\kappa}{\kappa}) - \chi_0\right) = \frac{1}{(\alpha - \beta)(\alpha - \gamma)}$$
$$\ln \frac{\left[\sqrt{(kb^2u_{4,1} + lb - 2ak - \sqrt{2}(kb)^{\frac{3}{2}}\beta)(\alpha - \gamma)} - \sqrt{(\alpha - \beta)(kb^2u_{4,1} + lb - 2ak - \sqrt{2}(kb)^{\frac{3}{2}}\gamma)}\right]^2}{|kb^2u_{4,1} + lb - 2ak - \sqrt{2}(kb)^{\frac{3}{2}}\alpha|}. \tag{22}$$

$$\pm \left( \sqrt{\frac{b}{2k}} (kx + ly + \mu \frac{t^{\kappa}}{\kappa}) - \chi_0 \right) = \frac{1}{(\alpha - \beta)(\alpha - \gamma)}$$

$$\ln \frac{[\sqrt{(k^2 b^2 v_{4,1} + l^2 b - 2alk - \sqrt{2}(kb)^{\frac{3}{2}} l\beta)(\alpha - \gamma)} - \sqrt{(\alpha - \beta)(k^2 b^2 v_{4,1} + l^2 b - 2alk - \sqrt{2}(kb)^{\frac{3}{2}} l\gamma)}]^2}{|k^2 b^2 v_{4,1} + l^2 b - 2alk - \sqrt{2}(kb)^{\frac{3}{2}} l\alpha|}. \tag{23}$$

(ii) If $\alpha > \beta$ and $w < \gamma$, or $\alpha < \gamma$ and $w < \beta$, the implicit traveling wave solutions of Equation (1) can be expressed as below:

$$\pm \left( \sqrt{\frac{b}{2k}} (kx + ly + \mu \frac{t^{\kappa}}{\kappa}) - \chi_0 \right) = \frac{1}{(\alpha - \beta)(\alpha - \gamma)}$$

$$\ln \frac{[\sqrt{(kb^2 u_{4,2} + lb - 2ak - \sqrt{2}(kb)^{\frac{3}{2}} \beta)(\gamma - \alpha)} - \sqrt{(\beta - \alpha)(kb^2 u_{4,2} + lb - 2ak - \sqrt{2}(kb)^{\frac{3}{2}} \gamma)}]^2}{|kb^2 u_{4,2} + lb - 2ak - \sqrt{2}(kb)^{\frac{3}{2}} \alpha|}. \tag{24}$$

$$\pm \left( \sqrt{\frac{b}{2k}} (kx + ly + \mu \frac{t^{\kappa}}{\kappa}) - \chi_0 \right) = \frac{1}{(\alpha - \beta)(\alpha - \gamma)}$$

$$\ln \frac{[\sqrt{(k^2 b^2 v_{4,2} + l^2 b - 2alk - \sqrt{2}(kb)^{\frac{3}{2}} l\beta)(\gamma - \alpha)} - \sqrt{(\beta - \alpha)(k^2 b^2 v_{4,2} + l^2 b - 2alk - \sqrt{2}(kb)^{\frac{3}{2}} l\gamma)}]^2}{|k^2 b^2 v_{4,2} + l^2 b - 2alk - \sqrt{2}(kb)^{\frac{3}{2}} l\alpha|}. \tag{25}$$

(iii) If $\beta > \alpha > \gamma$, the implicit traveling wave solutions of Equation (1) can be expressed as below:

$$\pm \left( \sqrt{\frac{b}{2k}} (kx + ly + \mu \frac{t^{\kappa}}{\kappa}) - \chi_0 \right) = \frac{1}{(\beta - \alpha)(\alpha - \gamma)}$$

$$\arcsin \frac{(kb^2 u_{4,3} + lb - 2ak - \sqrt{2}(kb)^{\frac{3}{2}} \beta)(\alpha - \gamma) + (\alpha - \beta)(kb^2 u_{4,3} + lb - 2ak - \sqrt{2}(kb)^{\frac{3}{2}} \gamma)}{|(kb^2 u_{4,3} + lb - 2ak - \sqrt{2}(kb)^{\frac{3}{2}} \alpha)(\beta - \gamma)|}. \tag{26}$$

$$\pm \left( \sqrt{\frac{b}{2k}} (kx + ly + \mu \frac{t^{\kappa}}{\kappa}) - \chi_0 \right) = \frac{1}{(\beta - \alpha)(\alpha - \gamma)}$$

$$\arcsin \frac{(k^2 b^2 v_{4,3} + l^2 b - 2alk - \sqrt{2}(kb)^{\frac{3}{2}} l\beta)(\alpha - \gamma) + (\alpha - \beta)(k^2 b^2 v_{4,3} + l^2 b - 2alk - \sqrt{2}(kb)^{\frac{3}{2}} l\gamma)}{|(k^2 b^2 v_{4,3} + l^2 b - 2alk - \sqrt{2}(kb)^{\frac{3}{2}} l\alpha)(\beta - \gamma)|}. \tag{27}$$

**Case 5:** $D_2 > 0, D_3 = 0, D_4 = 0, E_2 = 0, G(w) = (w - \alpha)^3 (w - \beta)$, where $\alpha, \beta$ are real numbers.

When $w > \alpha$ and $w > \beta$, or $w < \alpha$ and $w < \beta$, the traveling wave solutions of Equation (1) take the form:

$$u_5(x, y, t) = \frac{8\sqrt{2}(\alpha - \beta)k^{\frac{3}{2}}}{\sqrt{b}[(\beta - \alpha)^2 (\sqrt{b}(kx + ly + \mu \frac{t^{\kappa}}{\kappa}) - \sqrt{2k}\chi_0)^2 - 4]} + \frac{\sqrt{2}\alpha(kb)^{\frac{3}{2}} + 2ak - lb}{kb^2} \tag{28}$$

$$v_5(x, y, t) = \frac{8\sqrt{2}(\alpha - \beta)lk^{\frac{1}{2}}}{\sqrt{b}[(\beta - \alpha)^2 (\sqrt{b}(kx + ly + \mu \frac{t^{\kappa}}{\kappa}) - \sqrt{2k}\chi_0)^2 - 4]} + \frac{\sqrt{2}l\alpha(kb)^{\frac{3}{2}} + 2akl - l^2 b}{k^2 b^2} \tag{29}$$

**Case 6:** $D_4 = 0, D_2 D_3 < 0, G(w) = (w - \alpha)^2 \left[ (w - l_1)^2 + s_1^2 \right]$. The traveling wave solutions of Equation (1) take the form:

$$u_6(x,y,t) = \frac{2ak-lb}{kb^2} + \sqrt{\frac{2k}{b}} \frac{\left[e^{\pm\sqrt{(\alpha-l_1)^2+s_1^2}\left(\sqrt{\frac{b}{2k}}(kx+ly+\mu\frac{t^\kappa}{\kappa})-\chi_0\right)} - \gamma\right] + \sqrt{(\alpha-l_1)^2+s_1^2}(2-\gamma)}{\left[e^{\pm\sqrt{(\alpha-l_1)^2+s_1^2}\left(\sqrt{\frac{b}{2k}}(kx+ly+\mu\frac{t^\kappa}{\kappa})-\chi_0\right)} - \gamma\right]^2 - 1}, \tag{30}$$

$$v_6(x,y,t) = \frac{2akl-l^2b}{k^2b^2} + l\sqrt{\frac{2}{kb}} \frac{\left[e^{\pm\sqrt{(\alpha-l_1)^2+s_1^2}\left(\sqrt{\frac{b}{2k}}(kx+ly+\mu\frac{t^\kappa}{\kappa})-\chi_0\right)} - \gamma\right] + \sqrt{(\alpha-l_1)^2+s_1^2}(2-\gamma)}{\left[e^{\pm\sqrt{(\alpha-l_1)^2+s_1^2}\left(\sqrt{\frac{b}{2k}}(kx+ly+\mu\frac{t^\kappa}{\kappa})-\chi_0\right)} - \gamma\right]^2 - 1}, \tag{31}$$

where $\gamma = \frac{\alpha-2l_1}{\sqrt{(\alpha-l_1)^2+s_1^2}}$.

**Case 7:** $D_4 > 0, D_3 > 0, D_1 > 0, G(w) = (w-\alpha_1)(w-\alpha_2)(w-\alpha_3)(w-\alpha_4)$, in which $\alpha_1, \alpha_2, \alpha_3, \alpha_4$ is real number and $\alpha_1 > \alpha_2 > \alpha_3 > \alpha_4$.

When $w > \alpha_1$ or $w < \alpha_4$, the traveling wave solutions of Equation (1) take the form:

$$u_{7,1}(x,y,t) = \frac{2ak-lb}{kb^2}$$
$$+\sqrt{\frac{2k}{b}} \frac{\alpha_2(\alpha_1-\alpha_4)\,\text{sn}^2\left(\frac{\sqrt{(\alpha_1-\alpha_3)(\alpha_2-\alpha_4)}}{2}\left(\sqrt{\frac{b}{2k}}(kx+ly+\mu\frac{t^\kappa}{\kappa})-\chi_0\right),m\right) - \alpha_1(\alpha_2-\alpha_4)}{(\alpha_1-\alpha_4)\,\text{sn}^2\left(\frac{\sqrt{(\alpha_1-\alpha_3)(\alpha_2-\alpha_4)}}{2}\left(\sqrt{\frac{b}{2k}}(kx+ly+\mu\frac{t^\kappa}{\kappa})-\chi_0\right),m\right) - (\alpha_2-\alpha_4)},$$

$$v_{7,1}(x,y,t) = \frac{2akl-l^2b}{k^2b^2}$$
$$+l\sqrt{\frac{2}{kb}} \frac{\alpha_2(\alpha_1-\alpha_4)\,\text{sn}^2\left(\frac{\sqrt{(\alpha_1-\alpha_3)(\alpha_2-\alpha_4)}}{2}\left(\sqrt{\frac{b}{2k}}(kx+ly+\mu\frac{t^\kappa}{\kappa})-\chi_0\right),m\right) - \alpha_1(\alpha_2-\alpha_4)}{(\alpha_1-\alpha_4)\,\text{sn}^2\left(\frac{\sqrt{(\alpha_1-\alpha_3)(\alpha_2-\alpha_4)}}{2}\left(\sqrt{\frac{b}{2k}}(kx+ly+\mu\frac{t^\kappa}{\kappa})-\chi_0\right),m\right) - (\alpha_2-\alpha_4)},$$

$$(32)$$

$$u_{7,2}(x,y,t) = \frac{2ak-lb}{kb^2}$$
$$+\sqrt{\frac{2k}{b}} \frac{\alpha_4(\alpha_2-\alpha_3)\,\text{sn}^2\left(\frac{\sqrt{(\alpha_1-\alpha_3)(\alpha_2-\alpha_4)}}{2}\left(\sqrt{\frac{b}{2k}}(kx+ly+\mu\frac{t^\kappa}{\kappa})-\chi_0\right),m\right) - \alpha_3(\alpha_2-\alpha_4)}{(\alpha_2-\alpha_3)\,\text{sn}^2\left(\frac{\sqrt{(\alpha_1-\alpha_3)(\alpha_2-\alpha_4)}}{2}\left(\sqrt{\frac{b}{2k}}(kx+ly+\mu\frac{t^\kappa}{\kappa})-\chi_0\right),m\right) - (\alpha_2-\alpha_4)},$$

$$v_{7,2}(x,y,t) = \frac{2akl-l^2b}{k^2b^2}$$
$$+l\sqrt{\frac{2}{kb}} \frac{\alpha_4(\alpha_2-\alpha_3)\,\text{sn}^2\left(\frac{\sqrt{(\alpha_1-\alpha_3)(\alpha_2-\alpha_4)}}{2}\left(\sqrt{\frac{b}{2k}}(kx+ly+\mu\frac{t^\kappa}{\kappa})-\chi_0\right),m\right) - \alpha_3(\alpha_2-\alpha_4)}{(\alpha_2-\alpha_3)\,\text{sn}^2\left(\frac{\sqrt{(\alpha_1-\alpha_3)(\alpha_2-\alpha_4)}}{2}\left(\sqrt{\frac{b}{2k}}(kx+ly+\mu\frac{t^\kappa}{\kappa})-\chi_0\right),m\right) - (\alpha_2-\alpha_4)},$$

$$(33)$$

in which $m^2 = \frac{(\alpha_1-\alpha_4)(\alpha_2-\alpha_3)}{(\alpha_1-\alpha_3)(\alpha_2-\alpha_4)}$.

**Case 8:** $D_4 < 0, D_2D_3 \geqslant 0$, then $G(w) = (w-\alpha)(w-\beta)\left[(w-l_1)^2+s_1^2\right]$, where real number $\alpha > \beta, l_1, s_1 > 0$.

The traveling wave solutions of Equation (1) take the form:

$$u_8(x,y,t) = \frac{2ak-lb}{kb^2} + \sqrt{\frac{2k}{b}} \frac{\text{acn}\left(\frac{\sqrt{-2s_1m_1(\alpha-\beta)}}{2mm_1}\left(\frac{b}{2k}(kx+ly+\mu\frac{t^\kappa}{\kappa})-\chi_0\right),m\right) + e_2}{\text{ccn}\left(\frac{\sqrt{-2s_1m_1(\alpha-\beta)}}{2mm_1}\left(\frac{b}{2k}(kx+ly+\mu\frac{t^\kappa}{\kappa})-\chi_0\right),m\right) + e_4}, \tag{34}$$

$$v_8(x,y,t) = \frac{2akl - l^2b}{k^2b^2} + l\sqrt{\frac{2}{kb}}\frac{\text{acn}\left(\frac{\sqrt{-2s_1m_1(\alpha-\beta)}}{2mm_1}\left(\frac{b}{2k}(kx+ly+\mu\frac{t^\kappa}{\kappa})-\chi_0\right),m\right)+e_2}{\text{ccn}\left(\frac{\sqrt{-2s_1m_1(\alpha-\beta)}}{2mm_1}\left(\frac{b}{2k}(kx+ly+\mu\frac{t^\kappa}{\kappa})-\chi_0\right),m\right)+e_4}, \tag{35}$$

in which $e_1 = \frac{1}{2}(\alpha+\beta)e_3 - \frac{1}{2}(\alpha-\beta)e_4, e_2 = \frac{1}{2}(\alpha+\beta)e_4 - \frac{1}{2}(\alpha-\beta)e_3, e_3 = \alpha - l_1 - \frac{s_1}{m_1},$
$e_4 = \alpha - l_1 - s_1m_1, E = \frac{s_1^2+(\alpha-l_1)(\beta-l_1)}{s_1(\alpha-\beta)}, m_1 = E - \sqrt{E^2+1}, m^2 = \frac{1}{1+m_1^2}.$

**Case 9:** $D_4 > 0, D_2D_3 \leqslant 0$, then $G(w) = \left[(w-l_1)^2 + s_1^2\right]\left[(w-l_2)^2 + s_2^2\right]$, where $l_1, l_2, s_1, s_2$ are real and $s_1 \geqslant s_2 > 0$. The traveling wave solutions of Equation (1) take the form:

$$u_9(x,y,t) = \frac{2ak - lb}{kb^2}$$
$$+\sqrt{\frac{2k}{b}}\cdot\frac{e_1\,\text{sn}\left(\eta\left(\frac{b}{2k}(kx+ly+\mu\frac{t^\kappa}{\kappa})-\chi_0\right),m\right)+e_2\,\text{cn}\left(\eta\left(\frac{b}{2k}(kx+ly+\mu\frac{t^\kappa}{\kappa})-\chi_0\right),m\right)}{e_3\,\text{sn}\left(\eta\left(\frac{b}{2k}(kx+ly+\mu\frac{t^\kappa}{\kappa})-\chi_0\right),m\right)+e_4\,\text{cn}\left(\eta\left(\frac{b}{2k}(kx+ly+\mu\frac{t^\kappa}{\kappa})-\chi_0\right),m\right)}, \tag{36}$$

$$v_9(x,y,t) = \frac{2akl - l^2b}{k^2b^2}$$
$$+l\sqrt{\frac{2}{kb}}\cdot\frac{e_1\,\text{sn}\left(\eta\left(\frac{b}{2k}(kx+ly+\mu\frac{t^\kappa}{\kappa})-\chi_0\right),m\right)+e_2\,\text{cn}\left(\eta\left(\frac{b}{2k}(kx+ly+\mu\frac{t^\kappa}{\kappa})-\chi_0\right),m\right)}{e_3\,\text{sn}\left(\eta\left(\frac{b}{2k}(kx+ly+\mu\frac{t^\kappa}{\kappa})-\chi_0\right),m\right)+e_4\,\text{cn}\left(\eta\left(\frac{b}{2k}(kx+ly+\mu\frac{t^\kappa}{\kappa})-\chi_0\right),m\right)}, \tag{37}$$

in which $e_1 = l_1e_3 + s_1e_4, e_2 = l_1e_4 - s_1e_3, e_3 = -s_1 - \frac{s_2}{m_1}, e_4 = l_1 - l_2, E = \frac{(l_1-l_2)^2+s_1^2+s_2^2}{2s_1s_2},$
$m_1 = E + \sqrt{E^2-1}, m^2 = \frac{m_1^2-1}{m_1^2}, \eta = \frac{s_2\sqrt{(e_3^2+e_4^2)(m_1^2e_3^2+e_4^2)}}{e_3^2+e_4^2}.$

## 3. Numerical Simulation

In order to understand the dynamical processes and mechanisms of complex phenomena of the fractional coupled Konopelchenko–Dubrovsky model, numerical simulations of the obtained soliton solutions are given in this section. As is vividly shown in Figures 2a, 3a and 4a, $u_1(x,y,t), u_2(x,y,t)$ and $u_3(x,y,t)$ stand for the tangent function solutions, the rational function solutions and the hyperbolic function solutions, respectively. Figures 2b, 3b and 4b denote the level curve at time $t = 1$. Furthermore, Figures 2c, 3c and 4c represent the density plots. Figures 2d, 3d and 4d stand for the contour plots.

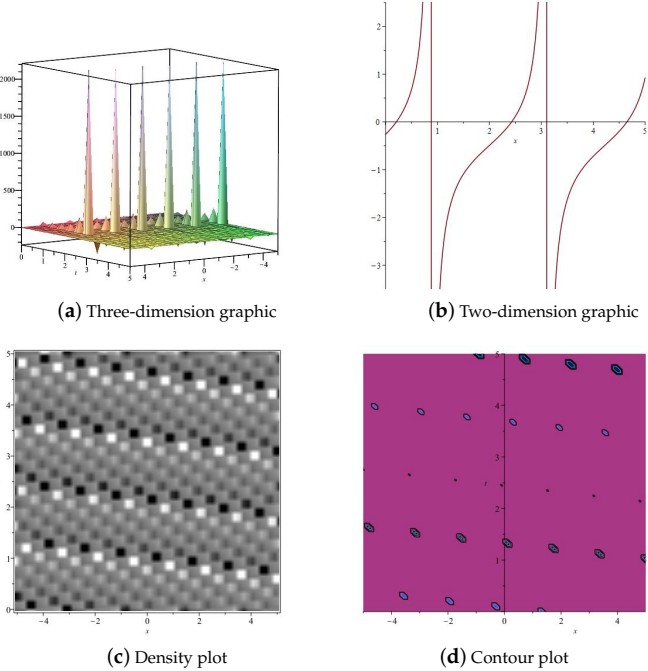

(**a**) Three-dimension graphic

(**b**) Two-dimension graphic

(**c**) Density plot

(**d**) Contour plot

**Figure 2.** Equation (14) for $a = 3, b = 4, k = 1, l = 2, \mu = 16$.

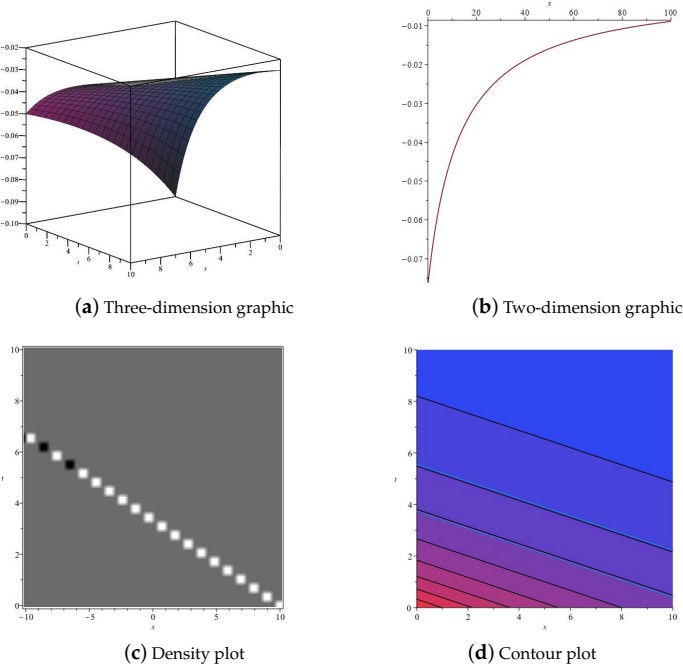

(**a**) Three-dimension graphic   (**b**) Two-dimension graphic

(**c**) Density plot   (**d**) Contour plot

**Figure 3.** Equation (16) for $a = k = l = 1, b = 2, \mu = 3$.

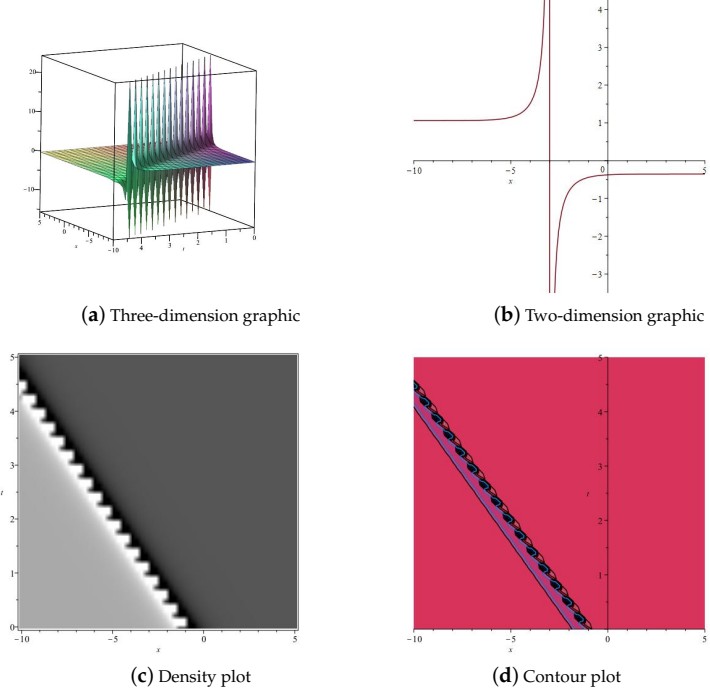

(**a**) Three-dimension graphic   (**b**) Two-dimension graphic

(**c**) Density plot   (**d**) Contour plot

**Figure 4.** Equation (18) for $a = k = l = 1, b = 2, \mu = 2, \alpha = \frac{1}{\sqrt{2}}, \beta = \frac{-1}{\sqrt{2}}$.

## 4. Conclusions

In this article, we have presented the traveling wave solutions of Equation (1) via the complete discriminant system method, which is one of the most useful tools in solving NLEEs. The trigonometric function solutions, the rational function solutions, the hyperbolic function solutions, the exponential function solutions, and the Jacobian elliptic function solutions are obtained. Lastly, in order to understand the mechanisms of physical phenomena for Equation (1), we have also depicted two-dimensional and three-dimensional diagrams. In future work, we will focus on the traveling wave solutions and dynamic behavior of

more complex NLEEs. Furthermore, we will also use the Darboux transformations to discuss the N-soliton solutions of more complex NLEEs.

**Author Contributions:** Software, Z.L.; Writing—original draft, J.W. All authors have read and agreed to the published version of the manuscript.

**Funding:** This research received no external funding.

**Data Availability Statement:** No new data were created or analyzed in this study. Data sharing is not applicable to this article.

**Conflicts of Interest:** The authors declare no conflicts of interest.

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
