# Peer review of "A Dynamical Analysis and New Traveling Wave Solution of the Fractional Coupled Konopelchenko–Dubrovsky Model"

_fractalfract, doi:10.3390/fractalfract8060341_

Round 1

Reviewer 1 Report

Comments and Suggestions for Authors

The manuscript exhibits encouraging prospects and may be suitable for publication subject to the execution of the subsequent recommended modifications.

1.     I suggest thoroughly revising the entire manuscript to address any grammatical

mistakes.

2.     Improve the introduction section and include the novelty of the work in the

introduction section of the paper with respect to other works.

3.     I suggest incorporating a conclusion section that includes insights into potential future work.

4.     The introduction can be improved by giving readers more background information about the partial fractional differential equations. Therefore, the following resources can be mentioned in the introduction which can be helpful for more related extensions or generalizations of the results in this paper in future research works: 10.1016/j.chaos.2022.111949, https://doi.org/10.1002/asjc.3389 

Comments on the Quality of English Language

No comments.

Author Response

The authors would like to express their great thankfulness to the reviewers for your much constructive, detailed and helpful advice regarding revising this manuscript.

  1. I suggest thoroughly revising the entire manuscript to address any grammatical

Answer 1 : Thank you for the reviewer's suggestions. We have carefully revised all grammar issues in this article.

  1. Improve the introduction section and include the novelty of the work in theintroduction section of the paper with respect to other works.

Answer 2 : Thank you for the reviewer's suggestions. We have added content to the introduction to make it more engaging for readers to read.

  1. 3.I suggest incorporating a conclusion section that includes insights into potential future work.

 Answer 3 : Thank you for the reviewer's suggestions. We have added a prospect for future work in the conclusion section.

4.The introduction can be improved by giving readers more background information about the partial fractional differential equations. Therefore, the following resources can be mentioned in the introduction which can be helpful for more related extensions or generalizations of the results in this paper in future research works:

10.1016/j.chaos.2022.111949, https://doi.org/10.1002/asjc.3389 

Answer 4 : Thank you for the reviewer's suggestions. We have carefully read the papers you mentioned and cited them in the references section.

Reviewer 2 Report

Comments and Suggestions for Authors

The authors construct traveling wave solutions to a fractional coupled Konopelchenko-Dubrovsky model. The adopted technique is the polynomial complete discriminant system method. There is another existing powerful approach to traveling wave solutions, called the transformed rational function method, on which the authors might like to offer some comparison. Note that N-soliton solutions of interesting nonlinear integrable flows involving reflection points of coordinates are generated in detail by Darboux transformations. If the authors could remark or comment on emergence of nonlinear dispersive wave phenomena in those nonlocal integrable nonlinear wave equations, the manuscript would become more beneficial to the interested audience. In summary, it is recommended that an enriched or revised version of the manuscript be accepted for publication in this journal.  

Comments on the Quality of English Language

In the abstract, "to study the traveling wave solution" should be "to study traveling wave solutions". "two-dimensional phase portrait is drawn. Finally, 3D-diagram and 2D-diagram" should be "a two-dimensional phase portrait is drawn. Finally, a 3D-diagram and a 2D-diagram".

Author Response

Response to Reviewer #2 Comments

  1. The authors construct traveling wave solutions to a fractional coupled Konopelchenko-Dubrovsky model. The adopted technique is the polynomial complete discriminant system method. There is another existing powerful approach to traveling wave solutions, called the transformed rational function method, on which the authors might like to offer some comparison. Note that N-soliton solutions of interesting nonlinear integrable flows involving reflection points of coordinates are generated in detail by Darboux transformations. If the authors could remark or comment on emergence of nonlinear dispersive wave phenomena in those nonlocal integrable nonlinear wave equations, the manuscript would become more beneficial to the interested audience. In summary, it is recommended that an enriched or revised version of the manuscript be accepted for publication in this journal.

Answer 1 :Thank you for the constructive comments and suggestions from the reviewer. The methods mentioned by the reviewer are indeed hot topics in current research on nonlinear partial differential equations. However, the methods mentioned by our reviewers are not familiar. Therefore, further research is needed. In this revision, we will still expand the content of the introduction and conclusion sections.
2. In the abstract, "to study the traveling wave solution" should be "to study traveling wave solutions". "two-dimensional phase portrait is drawn. Finally, 3D-diagram and 2D-diagram" should be "a two-dimensional phase portrait is drawn. Finally, a 3D-diagram and a 2D-diagram".

Answer 2 :Thank you for the reviewer's suggestions. There are indeed some printing errors in this article, and we will carefully revise it.

Reviewer 3 Report

Comments and Suggestions for Authors

As attached

Comments on the Quality of English Language

Needs rework

Author Response

Response to Reviewer #3 Comments

  1. The details of the basic motivation for the research should be properly stated.

Answer I : Thank you for the reviewer's suggestions. In the revised manuscript, in the introduction, we will further explain the research motivation of this article.

  1. There are a lot of grammar issues in the write-up. For instance, see the following sentence:

III. "In the present era, nonlinear evolution equations (NLEEs) are employed in a number of areas like physics, chemistry, biology, fluid dynamics, engineering, optical fiber, plasma and hydrodynamics." Should be:

  1. "In the present era, nonlinear evolution equations (NLEEs) are employed in numerous fields such as physics, chemistry, biology, fluid dynamics, engineering, optical fiber technology, plasma physics, and hydrodynamics."

Answer II,III,IV : Thank you to the reviewer. We will carefully revise the paper and resolutely eliminate the occurrence of problems here.

  1. Authors should ensure that all figures and diagrams are clearly labeled and referenced in the text.

Answer V : Thank you to the reviewer. We will add the content mentioned by the reviewer in the revised manuscript.

  1. State the practical applications of the generalized fractional calculus over the classical integer form

Answer VI: Thank you for the reviewer's suggestions. We will state the issue you mentioned in the introduction.